

# Seasonal and intra-diurnal variability of small-scale gravity waves in OH airglow at two Alpine stations

Patrick Hannawald[1], Carsten Schmidt[2], René Sedlak[1], Sabine Wüst[2], and Michael Bittner[1,2]

[1]University of Augsburg, Germany - Institute of Physics
[2]German Aerospace Center, Germany - German Remote Sensing Data Center

**Correspondence:** Patrick Hannawald (patrick.hannawald@physik.uni-augsburg.de)

**Abstract.**

Between December 2013 and August 2017 the instrument FAIM (Fast Airglow IMager) observed the OH airglow emission at two Alpine stations. One year of measurements was performed at Oberpfaffenhofen, Germany (48.09° N, 11.28° E) and two years at Sonnblick, Austria (47.05° N, 12.96° E). Both stations are part of the Network for the detection of mesospheric change (NDMC). The temporal resolution is two frames per second and the field of view is 55 km x 60 km and 75 km x 90 km at the OH layer altitude of 87 km with a spatial resolution of 200 m and 280 m per pixel, respectively. This results in two dense datasets allowing precise derivation of horizontal gravity wave parameters. The analysis is based on a two-dimensional Fast Fourier Transform with fully automatic peak extraction. By combining the information of consecutive images time-dependent parameters such as the horizontal phase speed are extracted. The instrument is mainly sensitive to high-frequency small- and medium-scale gravity waves. A clear seasonal dependency concerning the meridional propagation direction is found for these waves in summer in direction to the summer pole. The zonal direction of propagation is eastwards in summer and westwards in winter. Investigations of the data set revealed an intra-diurnal variability, which may be related to tides. The observed horizontal phase speed and the number of wave events per observation hour are higher in summer than in winter.

## 1 Introduction

Hydroxyl (OH) airglow, originally investigated by Meinel (1950), can be used as a tracer for atmospheric dynamics in the middle atmosphere, especially for the investigation of gravity waves (Peterson (1979), Taylor et al. (1993), Gardner and Taylor (1998) and many more). The OH airglow layer is located at about 87 km altitude and has a half width of roughly 4 km (Baker and Stair (1988)). Newer studies e.g. from von Savigny (2015) or Wüst et al. (2017) show that the altitude change can be up to a few kilometres, also the shape of the distribution with height may vary. Many OH bands contribute to the overall intensity in the visible and short wave infrared range (see e.g. Rousselot et al. (2000)). However, the intensity in the short wave infrared is much higher than in the visible range. Therefore, exposure times of instruments observing the OH airglow can be much lower when addressing the short wave infrared emissions. Thus, the temporal resolution of the FAIM data is comparatively high with up to two frames per second (Hannawald et al. (2016), Sedlak et al. (2016)).





Changes in pressure and temperature lead to intensity fluctuations in the OH layer. These perturbations can be measured and are most often caused by atmospheric gravity waves or other atmospheric wave types. Gravity wave parameters such as horizontal wavelength and observed phase speed can be derived from images of the OH airglow layer (see Peterson (1979), Taylor et al. (1995), Hecht et al. (2000), Nakamura et al. (1999), Mukherjee et al. (2010), Pautet et al. (2014) and others).

It is well known that gravity waves influence the circulation on a global scale (see Fritts and Alexander (2003) for an overview). The residual meridional circulation in the mesosphere is driven by breaking gravity waves (Holton (1983), Becker (2009)). In this context small-scale and short-period gravity waves are of particular interest (Fritts and Vincent (1987)).

Gravity waves are often generated in the troposphere, when large horizontal flows of air masses encounter obstacles and get displaced vertically. Mountain ridges or coastlines can serve as such an obstacle. In Europe, the Alps significantly influence

large scale flows in the troposphere. Dependent on the vertical wind structure, some of these waves can travel upward up to the mesosphere. In order to investigate the characteristics of gravity waves, a short-wave infrared imager has been deployed at two Alpine stations (belonging to the network for the detection of mesospheric change; https://ndmc.dlr.de) observing the OH airglow with high spatio-temporal resolution for over three years. This gives a rather large and dense dataset of OH airglow images which is investigated the first time in this study. The focus of the investigation is on small-scale gravity waves with

horizontal wavelengths smaller than 50 km.

In section 2 the instrument, the data basis, and data preprocessing is briefly described. Section 3 explains the analysis in detail including the derivation of the unambiguous propagation direction, phase speed and period of the waves. The results are presented in section 4 and discussed in section 5.

## 2   Instrumentation and data

The instrument FAIM 1 (Fast Airglow IMager) is based on a cooled InGaAs-photodiode array (256 px x 320 px) with a spectral sensitivity from about $0.9\,\mu$m to $1.65\,\mu$m. Due to the spectral intensity distribution of the OH airglow emission (e.g. Rousselot et al. (2000)) mainly the emissions of OH(3-1) and OH(4-2) contribute to the observed intensity (see e.g. Hannawald et al. (2016) for further information). Therefore, the possible acquisition time for the images is comparatively low allowing for a temporal resolution of two frames per second. Equipped with a narrow angle lens with a field of view (FOV) of $19.5°$ to $24.1°$

the spatial resolution is comparatively high. The observed area at 87 km is large enough to investigate both, small-scale gravity waves (up to about 50 km horizontal wavelength) as well as instability structures (down to horizontal scales of 200 m). Further details concerning the instrument, its FOV, and comparison to spectrometer data are given in Hannawald et al. (2016).

From 17th December 2013 to 26th January 2015 FAIM 1 was located at Oberpfaffenhofen (OPN), Germany ($48.09°$ N, $11.28°$ E) with a zenith angle of $45°$. The resulting average resolution at the height of the OH airglow layer is about $200\,$m px$^{-1}$

and the covered area is 55 km x 60 km.

From 3rd August 2015 to 26th July 2017 the instrument was located at Sonnblick Observatory (SBO), Austria ($47.05°$ N, $12.95°$ E). The zenith angle of $55°$ for the second configuration is higher and corresponds to a smaller spatial resolution and a larger observed area ($280\,$m px$^{-1}$ and 70 km x 95 km).





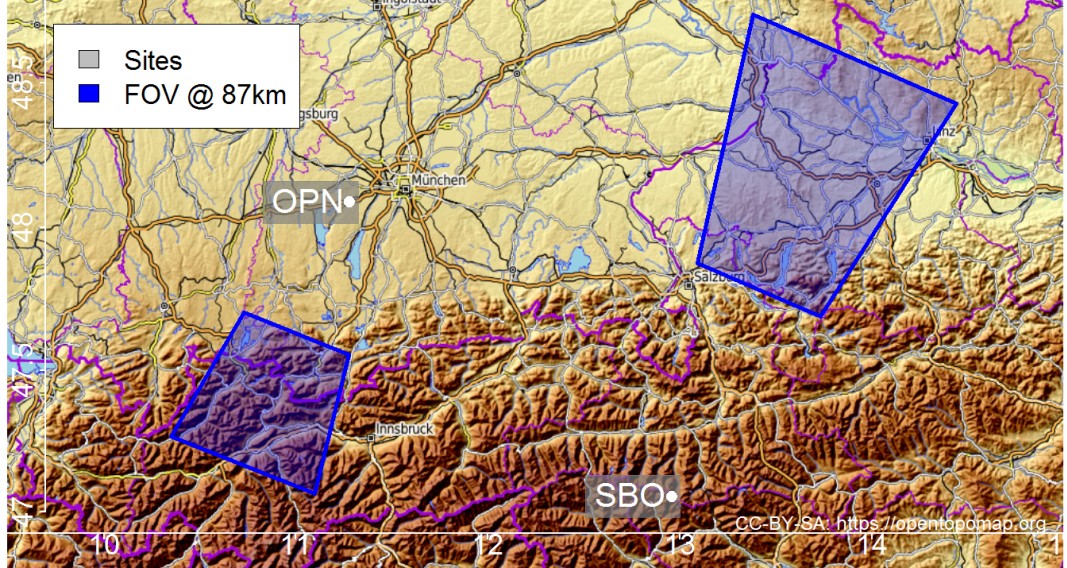

**Figure 1.** The two instrument sites in the Alpine region, Oberpfaffenhofen (OPN, 2013–2015) and Sonnblick Observatory (SBO, 2015–2017). Observed areas at 87 km altitude from OPN (left trapezium) and SBO (right trapezium) are drawn in blue. Both stations belong to the NDMC.

The position and size of the FOV within the Alps and at their foothills can be read from Fig. 1 where the left FOV corresponds to the site OPN and the right FOV to SBO. The OPN data are acquired above the mountains whereas the SBO data are acquired from the northern Alpine foothills.

For the whole observation period the integration time of the camera was set to 0.5 s. For several nights in the first quarter of
5 the year 2014 a long pass filter was used to limit the spectral sensitivity to 1.3–1.65 $\mu$m to exclude contributions from $O_2$(0-0) at 1.27 $\mu$m. In our later study (Hannawald et al. (2016)), we showed that the contribution of $O_2$ to the recorded signal is negligible. So, the filter is not required and was therefore removed during later observations. However, these modifications do not influence the desired wave parameters. For SBO the settings stayed unchanged for the whole time period. It should be noted that the entire analysis focusses on the intensity distribution within each image and not on seasonal or day-to-day changes of
10 airglow intensity.

The number of acquired images per month is depicted in Fig. 2. The black bars show the images with clear sky conditions which are used for analysis. The stacked grey bars correspond to images not analysed due to cloud cover or moon light visible in the FOV. The periods with good weather conditions were carefully selected by manually checking keograms and, where necessary, inspecting the image sequences of the respective time frames by eye.

For both stations each month is represented by more than a hundred thousand images (Fig. 2). The maximum number of images in one month is almost two million in December 2016. 7.2 million images (about 1000 hours of airglow observation)





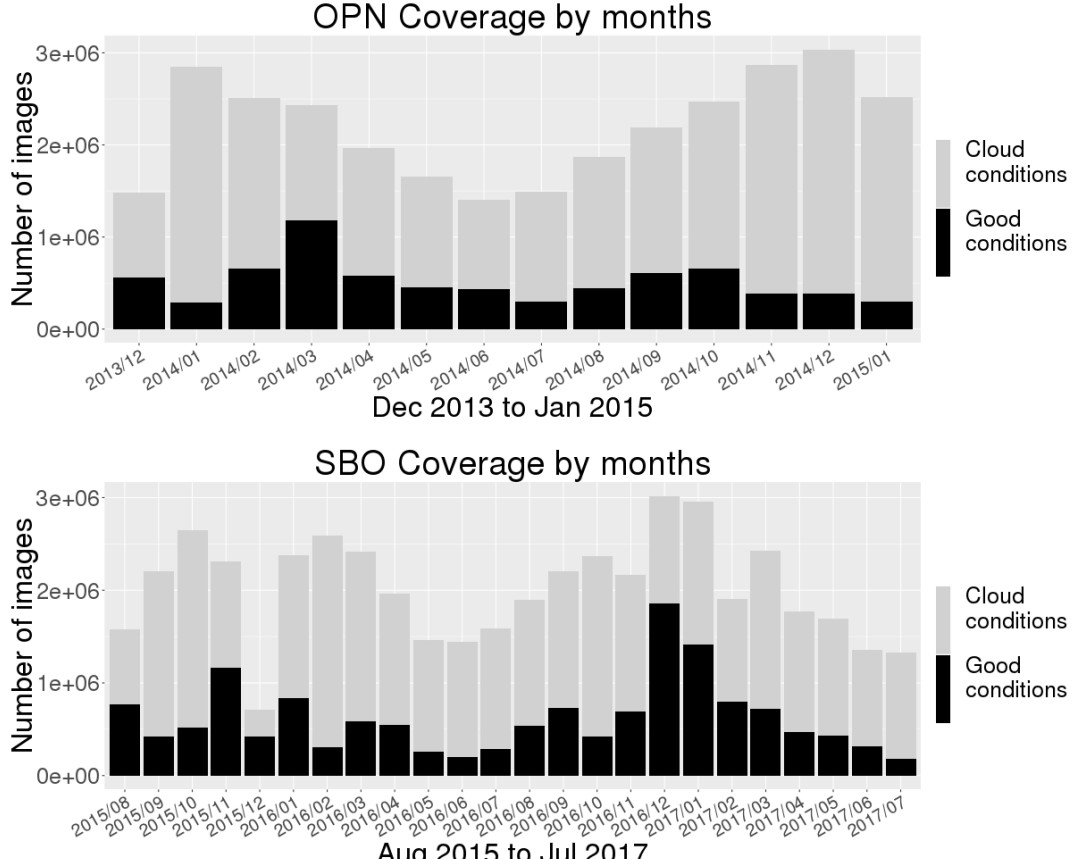

**Figure 2.** Overview of the data sets for both stations Oberpfaffenhofen (OPN) and Sonnblick Observatory (SBO). The coverage is grouped by months. The black bars show the number of images with good observation conditions which are used for analysis. The grey bars give the portion of images with bad conditions, e.g. cloud cover or intense moon light.

of data for OPN are analysed as well as 14.8 million images (about 2000 hours) for SBO. Overall, 28 % of all data are found to be good and are analysed.

The preprocessing of the images is explained in the following paragraphs. First, the images are flat fielded to correct them for fixed pattern noise and camera artefacts. This includes removing dead pixels with interpolation over neighbouring pixels.
5   Fig. 3 shows the steps of preprocessing for an exemplary image where (a) references to the flat fielded image. Additionally, a Gaussian blur is applied here to the image with a kernel size of three to reduce high frequency noise.

Image (b) shows the result of the star removal. Due to the high spatial resolution, stars usually cover several pixels (in individual cases up to 50 pixels/star!) and are present in every image showing airglow. Therefore, an optimal star removal requires a sophisticated approach, otherwise too many pixels will be modified. The star removal algorithm consists of three
10  steps:





**Figure 3.** Preprocessing steps explained for one image at 2016-05-26 01:52:41 UTC, for display reasons the contrast is adjusted for each image. (a) Flat-fielded image with Gaussian blur (kernel size 3) applied, (b) result of star removal, (c) image unwarped to equidistant grid and flipped around y-axis for satellite's view, (d) image cropped to square size, (e) middle section of point-symmetric FFT magnitude spectrum (red line: significant level, blue dots: identified maxima), (f) superposition of the four reconstructed plain waves, (g) the four plain waves side by side.

1. Identification of stars based on a threshold value: First a median blur is applied to image (a). The blurred image is subtracted from image (a). The stars as the high frequency parts of the image remain. All pixels with values above a pre-defined start threshold are considered to be stars. This normally identifies more than one, but not all pixels influenced by the star, so further characterisation is needed.

2. The characterisation of each star is performed by finding its center (denoted as star seed) and radius in order to get all pixels influenced by the star (denoted as star pixels): The maximum value of each star is searched and taken as the center of the star. The radius of the star is determined by looking at the four main directions from the star seed (top, bottom,



left, right). Starting at the star seed, the distance in each direction is increased individually for as long as the intensity value is lower than the former value in the respective direction. The maximum distance of the directions is taken as the star radius. All pixels in a circle around the star seed with the star radius are considered as star pixels.

3. All star pixels are interpolated by its nearest (non-star pixel) neighbours.

To overcome the arbitrary threshold in step 1, the steps 1 and 2 are repeatedly applied on the original image with a slightly increased or decreased threshold for each iteration. This is done for as long as the number of overall star pixels is between 2 and 7 % of all pixels. The final threshold is then used for the actual star removal.

The starless image (Fig. 3 (b)) is then unwarped to an equidistant grid resulting in a trapezium-shaped FOV. The unwarping depends on the zenith angle of 55° (45° for OPN) following the procedure described in Hannawald et al. (2016). Furthermore, the image is flipped around the vertical axis to get a satellite's view, which can be projected onto a map (East is on the right side of the image then). Figure 3 (c) shows the result of this step. The largest possible square-shaped area within the trapezium is taken from the unwarped image (c) in order to prevent directional differences due to the shape of the analysed area. The result is shown in Fig. 3 (d).

Both, mean intensity and linear trend (i.e. a plane calculated by a two-dimensional linear regression) are subtracted from Fig. 3 (d). The image is then multiplied with a Kaiser-Bessel window (alpha = 4, Kaiser and Schafer (1980)) so that the edges fade to zero. The Kaiser-Bessel window has been chosen for its high side lobe damping. Additionally, zero padding is applied to further improve and resolve the peaks of the following Fourier analysis more clearly.

## 3 Analysis

Due to the huge amount of data an automatic approach is necessary for the analysis. The two-dimensional fast Fourier transform (2D-FFT, or even 3D-FFT) is often used to extract wave parameters from airglow images (e.g. Gardner et al. (1996), Garcia et al. (1997), Coble et al. (1998), Matsuda et al. (2014) and others). The focus of this article is on the automatic analysis of the Fourier spectra, i.e. on identifying peaks, combining the information of consecutive images, grouping the results in wave events and investigating them individually.

The previously described preprocessing steps are performed for each image and the FFT is calculated subsequently. The resulting two dimensional complex spectra are then analysed for the major peaks. Figure 3 (e) shows the Fourier magnitude spectrum (Mag = $\sqrt{\mathrm{Re}^2 + \mathrm{Im}^2}$). The point of origin of $k_x$ and $k_y$ is at the center of the spectrum. In Fig. 3 (e) just the middle part of the spectrum is shown as the rest is not significant.

In order to identify the peaks in the Fourier spectrum a significance level is estimated. Accordingly, for each preprocessed image 100 random matrices based on the standard deviation of the preprocessed image are created and the 2D-FFT is applied to them in the same manner. For each of the 100 random spectra the magnitude is calculated and the respective maximum values are extracted. The 95 % percentile of these maxima is then taken as significance level. Values lower than the significance level (red contour in Fig 3 (e)) are excluded from further analysis.





Now, the peaks in the spectrum have to be identified and distinguished correctly. Therefore, a local maximum search is accomplished with the condition that the distance between two maxima has to be at least three pixels (5x5 kernel for maximum search). For all these local maxima at position $(x', y')$ the amplitude $A$ is determined by:

$$A = \sqrt{\mathrm{Re}(x', y')^2 + \mathrm{Im}(x', y')^2}. \tag{1}$$

The FFT reconstructs the image by superposing plain waves. If a wave signal is not adequately described by a plane wave (e.g. the wave crests and valleys are curved or show other irregularities) it will be composed of additional plane wave components. This leads to the fact that in some cases more than 10 different wave components are found in one image. Before further simplifying these cases, all signals are kept if they have at least 10 % of the intensity of the signal with the highest amplitude.

For visualization purposes Fig. 3 (f) shows the reconstruction of the original image based on the four (point-symmetric)
local maxima identified in the spectrum and their derived parameters (marked as blue dots in Fig. 3 (e)). (g) shows the four individual wave signatures as plain waves separately.

For each of these maxima, the wave parameters horizontal wavelength $\lambda$, angle of propagation $\alpha$ and Phase $\phi$ are determined:

$$\lambda = \frac{1}{\sqrt{k_x^2 + k_y^2}}, \tag{2}$$

$$\alpha = \arctan\left(\frac{k_y}{k_x}\right), \tag{3}$$

and

$$\phi = \arctan\left(\frac{\mathrm{Im}(x', y')}{\mathrm{Re}(x', y')}\right). \tag{4}$$

In order to bring together the signals extracted from each image individually, the wave signatures (waves with identical
wavelength and identical angle of propagation) are grouped into so-called wave events. It is assumed that a wave (band) or an instability feature with a wave-like appearance (ripple) will last for more than just a few seconds and should therefore be detected in several consecutive images, possibly with gaps of a few images. These groups of identical signatures in a given time interval should henceforth be denoted as wave events. A new signature is attributed to a previously identified wave event if it occurs less than 30 seconds after the last known signature of the event.

Each wave event with more than two occurrences of the respective wave signature (wave signature found in more than two images) is analysed in order to derive the overall time of occurrence, phase speed, period and unambiguous direction of propagation. Through linear regression the mean phase shift with time is determined from the phase information contained in the FFT. Phase jumps are considered in order to get the correct slope of the linear regression. The reciprocal absolute value of the slope gives the period $T$ of the wave event. With the phase shift with time $\dot{\varphi} = \frac{\Delta\varphi}{\Delta t}$ and the horizontal wavelength the
horizontal phase speed $v$ can be calculated:

$$v = \lambda \cdot |\dot{\varphi}|. \tag{5}$$





The sign of $\dot{\varphi}$ also provides the unambiguous direction of propagation.

Further statistical values are determined for each wave event such as the length of the time interval, in which the respective wave is observed, referred to as presence time of the event, and the number of occurrences within the presence time, which is an important indicator for the persistence of the wave event. These parameters are used as indicators to decide whether an

event can be considered as "important" wave. To be considered any further, an event has to be present for at least two minutes (240 images) in which the respective wave signature has to be found at least 100 times. The derived horizontal phase speed should be larger than $3\,\mathrm{m\,s^{-1}}$ and the residual standard error of the linear regression less than seven degree. These values were empirically derived from extensive testing.

Through this kind of filtering very small-scale waves or instability features (smaller than 5 to 10 km horizontal wavelength)

are under-represented as these signals change rapidly and are present for only a small amount of time. This has to be considered when interpreting the results. In this study, we focus on the more persistent wave events. Regarding the FOV, the side length of the analysed regions is 47 km for OPN and 61 km for SBO. FFT results with larger horizontal wavelengths (e.g. due to the wave being arranged along the diagonal line of the analysed region) are excluded from further interpretation.

## 4   Results

For investigating the results the datasets are split into summer (April to September) and winter (October to March) season. The predominant propagation direction in summer is similar for both stations (OPN and SBO) and is towards the North-East (NE) direction (Fig. 4). More than 46 % (OPN) and 55 % (SBO) of the waves propagate in this direction. During winter, the main propagation direction derived from OPN data is south-west (NE: 15 %, SE: 21 %, SW: 38 %, NW: 26 %). At SBO, the main propagation direction during winter is to north-west and south-west (NE: 16 %, SE: 22 %, SW: 27 %, NW: 36 %).

Integrated over summer and winter, about 48 % (OPN) and 26 % (SBO) of the waves have less than 15 km horizontal wavelength. As mentioned above, the data filtering process underestimates small-scale waves (Fig 4). Therefore, these small-scale waves are obviously quite persistent. Medium-scale waves with wavelengths from 15 km to two thirds of the respective FOV represent 34 % (OPN) and 55 % (SBO) of the detected waves. Larger-scale waves up to the side length of the FOV represent about 20 % of all waves.

For OPN, the amount of small-scale waves tends to be larger in winter with 51 % in comparison to 45 % in summer (within the above mentioned uncertainties caused due to the filter process). For medium- and large-scale waves (wavelengths larger than 15 km) the situation reverses (49 % during winter and 55 % during summer), however, this seasonal difference is quite low. At SBO, the situation is qualitatively similar. The occurrence rate is about 31 % during winter and 21 % during summer for the small-scale waves and 70 % during winter and 79 % during summer for the medium- and larger-scale waves.

The number of wave events normalised to the amount of available airglow observation hours by season at OPN shows a density of 6.1 events per hour during summer and 3.6 events per hour during winter. At SBO, the density is 7.1 events per hour in summer and 4.2 events per hour in winter. Thus, the density in winter is only 60% as high as in summer.





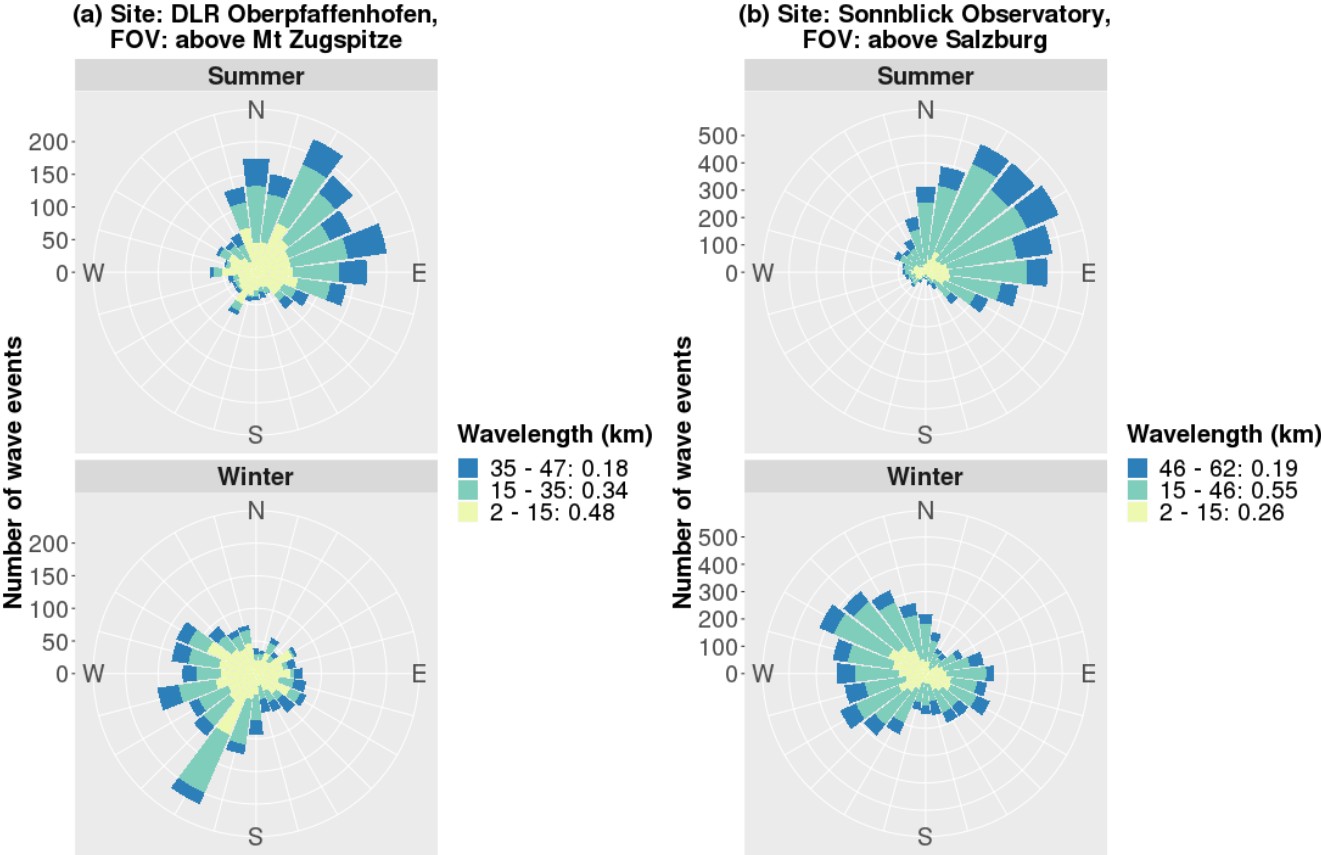

**Figure 4.** Directions of gravity wave propagation at Oberpfaffenhofen (a, 48.09° N, 11.28° E) and Sonnblick (b, 47.05° N, 12.95° E). The top (bottom) panels show the data for summer (winter) season. The colors refer to the horizontal wavelengths and are separated into small-scale waves (< 15 km), medium-scale waves (>= two thirds of the FOV: 35 and 46 km, respectively) and larger-scale waves (up to the side length of the respective FOV; 47 and 62 km, respectively). The numbers next to the wavelength legend represent the proportion of the respective bin.

In order to investigate the intra-diurnal variation of the direction of propagation, all wave events are binned according to the time of day of their occurrence (Fig. 5). The directions are grouped into the four quadrants NE, SE, SW, NW. The panels (a), (b), (e), and (f) show the distribution of wave events with time. Mainly due to the variation of the length of night, the maximum of wave events is around midnight (approx. 22 UTC to 1 UTC). The relative distribution of the different directions (Fig. 5 (c), (d), (g), (h)) reveals considerable intra-diurnal variations. Obviously, the propagation towards the North-East (NE) direction (red) is dominant with more than circa 40 % of the wave events for almost all hours (Fig. 5 (c) and (d)). In winter,



**Figure 5.** Number of gravity wave events and distribution of propagation directions as a function of time for OPN (left side) and SBO (right side). The top panels (a)–(d) refer to the summer season and the bottom panels (e)–(h) to the winter season. The data is grouped into North-East- (NE), South-East- (SE), South-West- (SW) and North-West- (NW) quadrants.





the SW direction (green) is prevalent at OPN and the NW and SW direction (blue and green) at SBO (Fig. 5 (g) and (h)) as already seen from Fig. 4. In general, there is a notable anti-correlation for opposing directions (NE to SW and NW to SE). However, the correlation coefficients underlie high uncertainties due to only few available data points and – more important – an unequal distribution of wave events which leads to an overestimation of the early and late night hours. The propagation

direction towards NE at SBO summer (Fig. 5 (d)), shows an oscillation-like pattern with a maximum at 23 UTC and minima at 17 UTC and 5 UTC which could be related to a period of about 12 hours. A shorter period oscillation may be seen at SBO winter NE direction (Fig. 5 (h)) with maxima at 1 UTC and 17 UTC and a minimum at 21 UTC. It would be related to an eight hour oscillation. However, these periods remain speculative, because the they are in the same range as the length of the night. This makes it impossible to investigate the exact values.

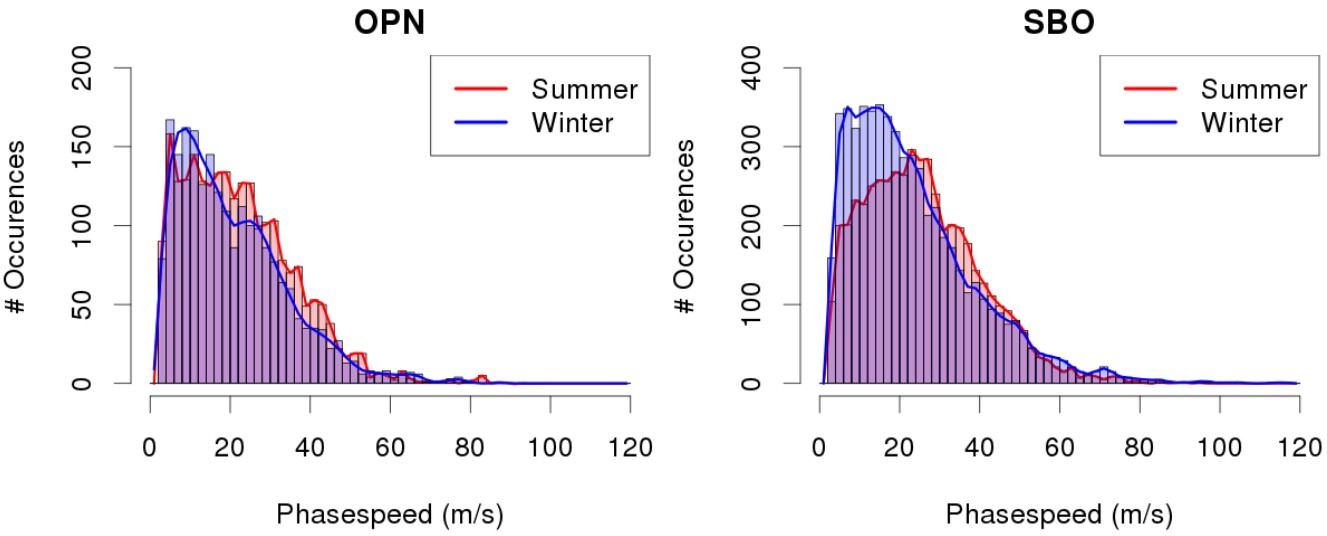

**Figure 6.** Absolute values of horizontal phase speed of the gravity wave events for OPN and SBO for summer and winter season in $\mathrm{m\,s^{-1}}$.

The distributions of observed horizontal phase speeds are shown in Figure 6 and 7. Figure 6 depicts the absolute values. During OPN winter, the maximum of the distribution is around $9\,\mathrm{m\,s^{-1}}$ with a secondary peak at about $25\,\mathrm{m\,s^{-1}}$. For OPN summer, the distribution is not as smooth as during winter (with peaks at about 5, 10, 18, 23, 30, 38, and $42\,\mathrm{m\,s^{-1}}$). The SBO distributions reveal peaks at 7 and $13\,\mathrm{m\,s^{-1}}$ in winter and 23 and $35\,\mathrm{m\,s^{-1}}$ in summer. The 95 % quantile of phase speed is 46 and $52\,\mathrm{m\,s^{-1}}$ considering both seasons and all directions, thus just a few of the observed wave events propagate faster than

that.

Table 1 shows the mean values and standard deviation of the phase speeds as a function of the direction. For both stations, the mean values are higher during summer than during winter with an increase of about 7 % (OPN, 22.2 to $20.7\,\mathrm{m\,s^{-1}}$) and 9 % (SBO, 25.5 to $23.5\,\mathrm{m\,s^{-1}}$). The observed horizontal phase speeds in SW direction during summer and NW direction during winter are significantly lower compared to the other directions (Table 1). Especially in summer, the phase speeds are higher in





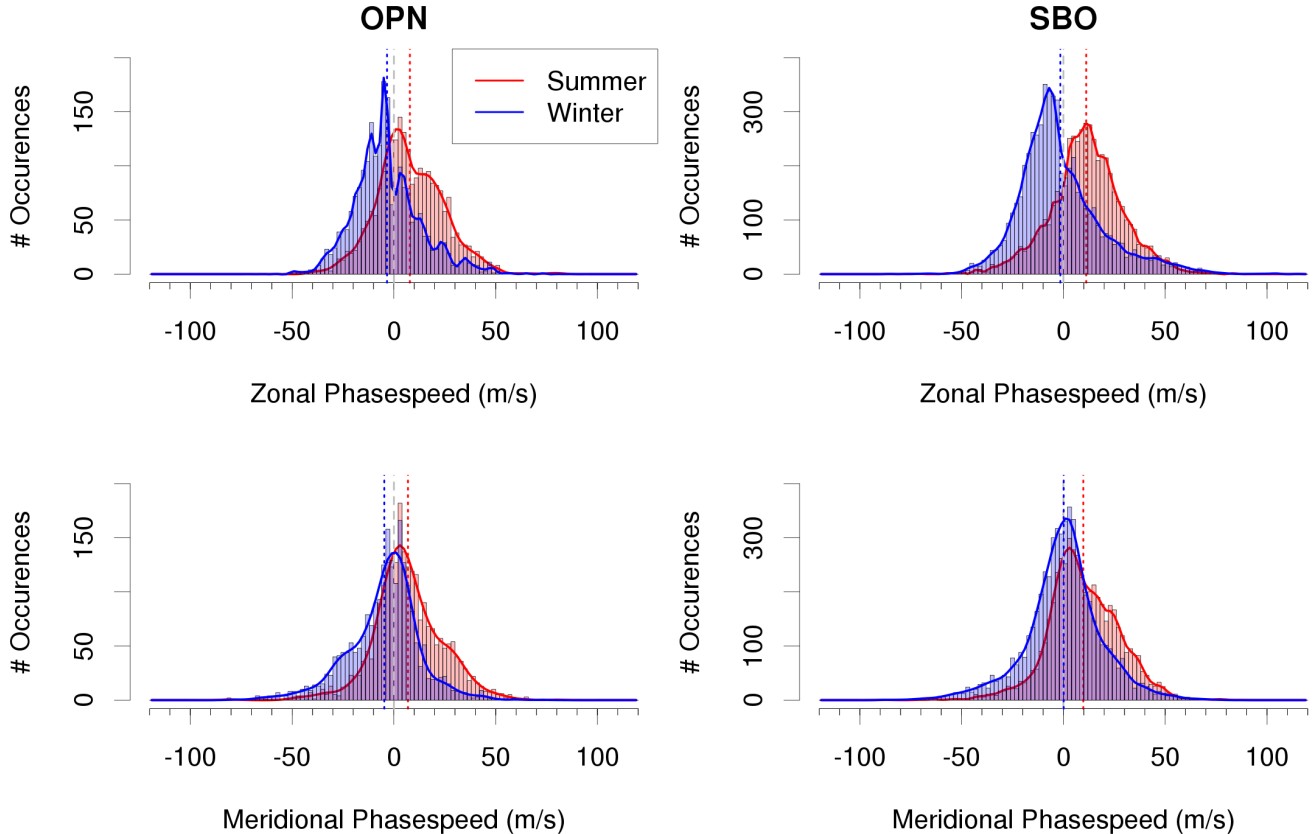

**Figure 7.** Horizontal phase speed of the gravity waves at OPN and SBO for summer and winter season in $\mathrm{m\,s^{-1}}$ separated by zonal and meridional components. Negative phase speed indicates westward or southward direction, positive phase speed eastward or northward direction. The grey dashed line marks zero phase speed and the red and blue dotted lines refer to the mean values over the respective summer and winter distribution, respectively.

**Table 1.** Mean (standard deviation) values of absolute horizontal phase speed in dependence of the station, season and direction.

| Station | Season | NE | SE | SW | NW | All |
|---------|--------|------|------|------|------|------|
| OPN | Summer | 24.4 (13.1) | 22.6 (14.2) | 17.4 (12.4) | 19.9 (14.1) | 22.2 (13.7) |
| OPN | Winter | 19.6 (13.5) | 23.9 (17.3) | 22.0 (12.4) | 17.0 (10.7) | 20.7 (13.6) |
| SBO | Summer | 26.3 (13.5) | 25.0 (15.6) | 17.9 (13.4) | 26.3 (14.6) | 25.5 (14.3) |
| SBO | Winter | 27.5 (17.5) | 27.7 (20.7) | 22.1 (13.5) | 20.2 (12.1) | 23.5 (15.9) |

eastward direction (NE and SE together) than in westward direction (SW and NW together). During winter at SBO it is similar, but for OPN winter, the behaviour is different (NE lower phase speed than SW).



**Table 2.** Statistical moments of horizontal phase speed distributions for station, season and direction. Negative phase speed indicates westward or southward direction, positive phase speed eastward or northward direction. The units of mean, median, standard deviation and peak are $\mathrm{m\,s^{-1}}$, the skewness is unitless. The positions of the peaks are determined on basis of the splines overlying the histograms in Fig. 7.

| Station | Season | Direction | Mean | Median | Std.Dev. | Skewness | Peak |
|---|---|---|---|---|---|---|---|
| OPN | Summer | Zonal | 7.8 | 6.2 | 16.6 | 0.32 | 1 |
| OPN | Summer | Meridional | 6.9 | 5.1 | 17.2 | 0.01 | 3 |
| OPN | Winter | Zonal | -3.4 | -4.9 | 16.3 | 0.61 | -5 |
| OPN | Winter | Meridional | -4.7 | -2.8 | 17.7 | -0.37 | 1 |
| SBO | Summer | Zonal | 11.2 | 11.4 | 18.0 | -0.09 | 11 |
| SBO | Summer | Meridional | 9.8 | 8.3 | 17.6 | -0.03 | 3 |
| SBO | Winter | Zonal | -1.5 | -5.1 | 20.9 | 0.95 | -7 |
| SBO | Winter | Meridional | 0.1 | 0.8 | 19.1 | -0.24 | 1 |

Figure 7 shows the distributions separated into zonal and meridional phase speeds. The dashed lines give the mean values of the distributions. For zonal and meridional phase speeds, respectively, and for both stations, the mean values are higher in summer than in winter. The mean values of all SBO distributions are higher (more toward positive numbers) than the respective equivalents of OPN distributions. Table 2 gives additional information about the distributions. These are approximately sym-
metric with skewness values smaller than 0.37 except for the zonal directions in winter where the distributions are right-skewed with values 0.61 and 0.95 (the standard error calculated with $\sqrt{\frac{6}{N}}$ (Press et al. (2007)) is 0.04 (OPN) and 0.02 (SBO)). The standard deviation at SBO is higher than for OPN. For the zonal direction in winter this difference is highest with 28 % increase (20.9 (SBO) to 16.3 (OPN)).

The observed wave periods are shown in Figure 8. The 10 % and 90 % quantiles of the distribution of periods range from
about 6 min to 50 min (OPN) and from 7 min to 70 min (SBO). More than 60 % of the wave events have periods between 10 and 60 min. Around 20 % of the waves show periods close to the Brunt-Väisälä-Period with 5 up to 10 min. A few events have periods smaller than 5 min which could be related to acoustic gravity waves. However, the investigation of these events is beyond the scope of this study. The distributions of periods have maxima at about 8 and 10 min at OPN for summer and winter, respectively. At SBO, the maxima are at 13 and 7 min. The distributions are highly right-skewed with values 3.3 and
3.1 at OPN in summer and winter, and values of 3.3 and 2.8 at SBO.

The main results are summarised in the following list:

- The main zonal propagation direction is eastwards during summer and westwards during winter. The main meridional propagation direction is northwards during summer. During winter, the meridional propagation direction is southwards at OPN and as well northwards as southwards at SBO (Fig. 4).



**Figure 8.** Top: Same as Fig 4, but for the observed wave periods. Bottom: Histograms of observed wave period for summer and winter season.

- We found an intra-diurnal variability of the propagation directions. The opposing directions seem to be anti-correlated (NE to SW and NW to SE). Oscillations with periods of the order of 8 to 12 h can be estimated. However, these values are speculative (Fig. 5).

- The number of wave events per observation hour, the means of zonal and meridional phase speeds, and the means of absolute horizontal phase speeds are higher in summer than in winter (Fig. 6, 7, and 8).



## 5  Discussion

In order to understand the results it is essential to know which part of the gravity wave spectrum is actually observed by our instruments. The most important restraints for the observations are the OH layer thickness and the FOV of the instruments. The former is for example discussed in Gardner and Taylor (1998) who argue that the observed vertical wavelengths have

to be larger than the OH layer width. Wüst et al. (2016) showed in their Figure 9c the reduction of the observed amplitude of a wave depending on the vertical wavelength for different values of the OH layer thickness. The horizontal wavelength is limited by the FOV (OPN: 47 km, SBO: 61 km). Figure 8 indicates that most periods are smaller than one hour. Conclusively, the waves contained in the dataset are referred to small horizontal wavelength, high-frequency gravity waves which are known to be important for momentum transport (see Zhang et al. (2014)) and which are of major importance for the mesospheric

circulation (Garcia and Solomon (1985), Fritts and Vincent (1987)).

The propagation directions of the gravity waves show a clear pattern of seasonal dependency. This behaviour for mesospheric gravity waves is not limited to the two Alpine stations OPN and SBO, but it is well known. For example Tang et al. (2014) and Vargas et al. (2015) compared several airglow observations of many research groups around the globe and find a meridional propagation towards the summer pole for many stations. The zonal component of the eastward propagation during summer and

westward propagation during winter is also dominant at many stations.

The seasonal variation of the zonal propagation direction can be explained by zonal stratospheric wind fields when assuming that the observed waves originate from lower atmospheric layers or are directly influenced by waves from lower layers for example by wave-wave interactions. There is a strong westward wind in summer and eastward wind in winter (Fleming et al. (1990)) filtering gravity waves which propagate with a lower speed in the same zonal direction. Thus, mostly eastward prop-

agating gravity waves will be observed in summer and westward propagating gravity waves in winter. This is also confirmed for example by Taylor et al. (1993) and McLandress (1998).

In winter, waves with positive zonal phase speed should consequently be generated in situ or above the stratospheric jet (e.g. by wind shear), propagate from above down to the airglow layer, or pass the stratospheric jet when it is unusually weak. The stratospheric wind filtering could also explain the skewness of the zonal phase speed distributions, because the filtering

is mainly affecting the slower eastward propagating waves in winter which are less likely to be observed in the mesosphere. Therefore, there is a bump in the distribution at these phase speeds.

The seasonal variation of the meridional propagation direction (Fig. 4) for summer season towards the North, for winter towards the South at OPN and towards the South and the North at SBO could be due to the meridional circulation in the mesosphere. According to Yuan et al. (2008) showing Na lidar data and global circulation model runs, the meridional circulation

reverses in summer and winter (in summer: to the South, in winter: to the North); additionally, it is much weaker in winter than in summer. Therefore, the filtering effect can be regarded as being stronger during summer than during winter. This could be an indication for the southward propagating summer waves to be more influenced and filtered out by the stronger meridional circulation while the winter waves are less influenced by the weaker meridional circulation and do not suffer such a strong filtering effect. This would explain the clear pattern during summer and the more arbitrary meridional propagation during





winter, especially at SBO. However, there are also other suggestions for the meridional preferential direction as discussed e.g. in Vargas et al. (2015).

At Oberpfaffenhofen, horizontal wavelengths and phase speeds were already derived in the years 2011 and 2015 (Wachter et al. (2015): time period from February to July 2011; Wüst et al. (2018): time period July to November 2015; our data
were acquired at SBO during the second time period). In these cases, a combination of three spectrometers and one scanning spectrometer, respectively, were used instead of an imaging system. The spectrometers are sensitive to larger horizontal wavelengths, which are related to a higher possible maximum of intrinsic phase speed (compare e.g. Fritts and Alexander (2003)). This could explain our phase speeds which are lower than in the above mentioned studies.

We find the wave event density in winter to be only 60 % of that in summer. Results from Tang et al. (2014) coincide with
our results. An explanation for this behaviour could be that the typical altitude for gravity wave breaking is lower in winter than in summer (Holton and Alexander (2000)). This would decrease the number of observed wave events in the airglow layer in winter.

The mean value of horizontal wavelengths in Tang et al. (2014) is roughly 35 km which is larger than presented here with 20 km (OPN) and 28 km (SBO) and the distribution of periods they determined has a peak around ten minutes and a high right-
skewness. The peak at the periods we determined is in the same range with 7 to 13 min. The observed phase speed of Tang et al. (2014) has a peak at 50 m s$^{-1}$ and is therefore much higher than in our observations with 22 m s$^{-1}$ (OPN) and 24 m s$^{-1}$ (SBO). Tang et al. (2014) find a major propagation direction to the South in winter which we could determine just for OPN winter. The differences of the stations OPN and SBO could be due to geographical or time-conditioned differences. The former difference could be induced by the underlying orography (the Alps in our measurements), the latter one due to the change of
prevailing wind structures. It is interesting to note here that the FOV of SBO is located at the foothills of the Alps (compare Fig. 1). One might suggest here a link to the northward propagating gravity waves in SBO winter which is not present at the other station OPN within the Alpine region.

Intra-diurnal variation can for example be induced by atmospheric tides which change the direction and absolute value of the wind vector within the night and exhibit a period of about 24 h, 12 h, and 8 h. However, the 12 h solar tide shows the largest
change, at least in the zonal wind (Sandford et al. (2006)). In order to investigate the possible influence of such phenomena, the relationship between time of day and the direction is determined (see Fig. 5). We find an anti-correlation between the opposing directions (NE and SW, NW and SE). This could be an indication for a tide filtering out waves propagating in one direction and therefore preferring the opposing direction. We also tried to estimate the periods of oscillations visible in the intra-diurnal variability (Fig. 5: 12 h for the NE direction at SBO summer as well as 8 h for NE at SBO winter). From theory and observations
(e.g. Oberheide et al. (2003); Silber et al. (2017)), we know that the diurnal and semi-diurnal tide are supposed to have the strongest amplitudes, however, also the ter-diurnal tide is prominent. Our observations fit to the mentioned studies, but due to the data limited to about 9 h in summer and 13 h in winter and the expected periods to be in this range or larger, this result is hypothetical. A more detailed analysis with highly resolved wind data for the Alpine region is needed to confirm this, but it is beyond the scope of this paper.



## 6 Summary

We showed two airglow observation data sets with high spatio-temporal resolution. The instrument FAIM was located in the Alpine region first at Oberpfaffenhofen, Germany (OPN, 48.09° N, 11.28° E) and then at Sonnblick Observatory, Austria (SBO, 47.05° N, 12.96° E). The preprocessing as well as the analysis technique based on the two-dimensional Fast Fourier Transform

with automatic peak extraction and grouping into wave events are explained extensively. Combining the phase information of consecutive images allows the derivation of additional parameters related to time, especially horizontal phase speed and wave period. In general, observing the OH airglow layer with our imager allows us to characterise the spectrum of high-frequency gravity waves. The horizontal propagation directions of gravity waves show a clear seasonal dependency to NE in summer. In winter, they are to SW at OPN and to SW/NW at SBO. The zonal directions can be well explained by stratospheric wind

filtering while the meridional propagation towards the summer pole (OPN and SBO summer) is not yet completely understood. We suggest the meridional circulation itself to be the reason for the meridional preferential direction which is faster in summer when we observe a stronger filtering than in winter. We assume the generally lower height of gravity wave breaking in winter to be the reason that the gravity wave event density in winter is just 60% of that in summer. Concerning the observed horizontal phase speeds we find 7–9 % higher phase speeds in summer than in winter. The mean phase speeds are $22\,\mathrm{m\,s^{-1}}$ at OPN and

$24\,\mathrm{m\,s^{-1}}$ at SBO. Very few events with absolute phase speeds higher than $50\,\mathrm{m\,s^{-1}}$ were found. The intra-diurnal variability is investigated by grouping the gravity waves according to their occurrence time within the night. We find an anti-correlation between opposing directions (NE and SW, NW and SE). A rough estimation of periods of $8\,\mathrm{h}$ and $12\,\mathrm{h}$ could be a hint at the influence of mesospheric tides on gravity wave propagation.

*Data availability.* The investigated data is archived at WDC-RSAT (World Data Center for Remote Sensing of the Atmosphere).

*Author contributions.* The conceptualisation of the project, the funding acquisition as well as the administration and supervision was done by Michael Bittner and Sabine Wüst. The operability of the instrument was assured by René Sedlak, Carsten Schmidt and Patrick Hannawald. Fruitful discussions between all authors led to significant improvements of the algorithms and results. The software for the analyses and the visualisation as well as the original draft was written by Patrick Hannawald. All co-authors essentially contributed to the investigation process and reviewed the draft carefully.

*Competing interests.* The authors declare that they have no conflict of interest.



*Acknowledgements.* This work received funding from the Bavarian State Ministry of the Environment and Consumer Protection by grant number TUS01UFS-67093 and grant number TKP01KPB-70581. The observations are part of the NDMC (Network for the Detection of Mesospheric Change; https://ndmc.dlr.de).



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
