# Peer review of "Seasonal and intra-diurnal variability of small-scale gravity waves in OH airglow at two Alpine stations"

_Atmospheric Measurement Techniques, 2018_

## Referee Comment (RC1) · Anonymous Referee #1 · 18 Oct 2018

General Comments The authors describe a method of calculating gravity wave parameters from sequences of images of OH nightglow emissions recorded by a very sensitive infrared imager. The field-of-view of the camera and the cadence of the images (two per second) is such that the instrument is sensitive mostly to high-frequency small- and medium-scale gravity waves. The algorithms used to derive horizontal wavelength, propagation direction and phase-speed of the waves from the image sequences are described in detail. The instrument was deployed at two sites at which different zenith angles were used. Results from each site are presented and analysed by season (summer or winter) and by time of the day (denoted as "intra-diurnal").

The manuscript is well organised and the data is clearly presented with consistent labelling and logical colour coding. The methods used to identify the gravity waves in the image sequences are correct and the description of the methods used are clear. The text includes an appropriate set of references. The results are specific to the two Alpine stations and are a valuable contribution to this field of study. The work is suitable for publication in AMT, provided that the minor points highlighted below are addressed.

Specific Comments The predominant propagation directions at both observing sites showed similar seasonal patterns. Meridional propagation was towards the pole during the summer, while zonal propagation was eastwards in summer and westwards in winter. The observed zonal propagation pattern is consistent with stratospheric wind filtering as reported by previous work in this field. Observed horizontal phase-speeds were higher in summer than in winter as were the number of "wave events" when normalised to observing time.

The diurnal variation in the number of events and the direction of propagation at one station in a particular direction (NE at SBO both summer and winter) followed a pattern which prompted the authors to suggest a relation to tidal periods. This latter point is very weak, and while the authors refer to it as "hypothetical" (page 16, line 33), the data presented (NE direction in Figure 5d) does not yield a 12 h tide as claimed on line 6 of page 11. This point needs to be clarified or the claim of a 12 h tide (mentioned in several places in the manuscript) should be omitted from the manuscript.

Technical corrections Page 1, Title; Is it necessary to include the word "intra"? Perhaps the reason for the use of "intra" is that the measurements do not cover the complete diurnal cycle at any time of the year. Page 1, Abstract, line 5; replace "results" by "resulted". Page 1, Abstract, line 8; insert a comma after "images". Page 1, Abstract, line 9; insert a hyphen between "phase" and "speed". This should be used consistently throughout the manuscript, e.g., line 13 of the abstract etc. Page 1, Abstract, line 11; use " .. in the direction of the summer pole. " instead of " in direction to the summer pole." Page 1, Abstract, line 13; insert a hyphen between "observation and "hour". Page

1, line 17: use "... has a half-width (full width at half maximum) of roughly 8 km" instead of "has a half width of roughly 4 km". Page 1, line 20 and all other occurrences; use "short-wave" instead of "short wave". Use this consistently; see e.g., page 2, line 11, etc. Page 2, line 14; insert "for" after "investigated". Page 2, line 20; clarify the meaning of px. Page 2, line 24; use "field-of-view" instead of "field of view". Page 2, line 28; insert a comma after "2015". Page 2, line 32/33; " ... corresponds to a larger observed area and a lower spatial resolution (70 km x 95 km and 280 m px-1 respectively)." instead of " ... corresponds to a smaller spatial resolution and a larger observed area (280 m px-1 and 70 km x 95 km).". Page 3, line 5; insert a comma after "2014". Page 4, line 6; explain "Gaussian blur ... with a kernel size of three ... " or provide a reference here. Page 6, line 3; " ... within the star radius ... " instead of " ... with the star radius ...". Page 6, line 16; use "zero-padding" instead of "zero padding" Page 7, line 12; use "... and phase ïAe are determined as follows:" instead of "... and Phase ïAe are determined:" Page 7, line 23; use " are henceforth denoted as wave events." instead of "... should henceforth be denoted as wave events." Page 7, line 24; do you mean "if it occurs more than 30 seconds after the last known signature of the event."? Page 8, line 5; insert "an" before "important". Page 8, line 7; use "seven degrees" instead of "seven degree". Page 8, line 9; insert a comma after "filtering". Page 11, line 6; the minima occur at 18 UTC and 3 UTC in the NE direction at SBO summer in Figure 5(d)" not at 17 UTC and 5 UTC as stated in the text. This raises a question of the inference of a 12 hour period. Page 11, line 8; omit "the" after "because". Page 11, Figure 6, yaxis label; correct the spelling of "occurrences"; x-axis label; use "Phase speed (m/s)" instead of "Phasespeed (m/s)" Page 12, Table 1 title; replace "in dependence of" by "as a function of"; Page 13, line 12; use "acoustic-gravity waves" instead of "acoustic gravity waves". Page 13, line 19; use " ... and northwards as well as southwards at SBO ... " instead of " ... and as well northwards as southwards at SBO ... ". Page 15, line 3; use "constraints on the observations" instead of "restraints for the observations". Page 15, line 7; replace "Conclusively," by "Therefore, Page 15, line 18; insert "in the stratosphere" before "(Fleming et al.". Page 16, line 29; The claim of a 12 h period for

СЗ

the NE direction at SBO summer is not supported by the data shown in Figure 5(d)

---

## Referee Comment (RC2) · Anonymous Referee #2 · 17 Nov 2018

General comments:

The current manuscript presents a well written and interesting study on the characterization of gravity wave parameters estimated from ground-based imaging observations of the OH-airglow near the mesopause. The underlying measurements cover a period of about three years and were performed in the central European Alpine region. The data processing and data analysis are described in detail. The results are interesting and to a large extent confirm earlier studies. The obtained results are not spectacularly new, but the manuscript deserves to be published, in my opinion. I only have some minor suggestions.

[Figure]

Specific comments:

Page 1, line 8: "By combining the information of consecutive images". I suggest adding a comma after this part of the sentence to enhance readability

Page 1, line 22: "Thus, the temporal resolution of the FAIM data is comparatively high" You haven't mentioned yet, what OH bands (i.e. what spectral range) FAIM observes. Perhaps this should be mentioned, otherwise the reader cannot tell, whether the statement makes sense.

Page 2, bottom sentences: you give the "average" spatial resolutions for the measurements at the two different sites. It's perhaps interesting for the reader to see, how the spatial resolution varies across the FOV and how it differs between the two different axes.

Page 4, line 5: "where (a) references to the". Perhaps better "where (a) corresponds to the" ?

Page 7, line 2: "that the distance between two maxima has to be at least three pixels (5x5 kernel for maximum search)".

Can you explain in a bit more details what "5x5 kernel for maximum search" exactly means in this context? This it not entirely clear, at least to me.

Page 12, Figure 7, ordinate labels: "Occurences" -> "Occurrences". Same applies to Figure 6

Page 13, line 6: "the standard error calculated with"

I don't fully understand what this refers to here? The standard error of what quantity? Is the following mathematical expression correct, i.e. SQRT(6/N) ?

Page 13, line 12: "which could be related to acoustic gravity waves".

These events could also correspond to Doppler-shifted GWs.

[Figure]

Page 15, bottom paragraph: it would be good to mention what typical meridional wind speeds in summer and winter are. I imagine that the daily mean meridional wind speed associated with the mesospheric residual circulation is quite small. However, tidal variations may be quite large.

Page 16, line 1: Please briefly discuss the suggestions by Vargas et al. (2015).

Page 16, line 13 to 18: Please mention that latitude of the Tang et al. observations.

---

## Author Comment (AC1) · 12 Dec 2018

Reply to Anonymous Referee 1: We thank Anonymous Referee 1 for the valuable comments from 18th Oct 2018.

Each comment is considered in the following paragraphs. All changes of the initial manuscript (based on comments of both Anonymous Referees) are tracked in the supplemental material.

Concerning the comment on the 12h tide, which we claimed to be present in Figure 5d) NE direction: Although a harmonic fit reveals oscillations on the order of 12 hours,

its uncertainty is rather high and therefore not reliable (it could as well as be 11 or 13 hours). So we decided to omit the specific numbers (also the claim of an 8h tide) from the manuscript (page/line: 11/6, 14/2, 16/29, 17/17). Technical corrections (page/line): Page 1, Title; Is it necessary to include the word "intra"? Perhaps the reason for the use of "intra" is that the measurements do not cover the complete diurnal cycle at any time of the year. - We decided to keep the "intra" to emphasize that it's not the day-to-day variation, but the variation within a day by the (available) hours.

Page 1, Abstract, line 5; replace "results" by "resulted". - Done.

Page 1, Abstract, line 8; insert a comma after "images". - Done.

Page 1, Abstract, line 9; insert a hyphen between "phase" and "speed". This should be used consistently throughout the manuscript, e.g., line 13 of the abstract etc. - Hyphen added to all instances of "phase-speed" (also Fig 6 and 7).

Page 1, Abstract, line 11; use "... in the direction of the summer pole. " instead of " in direction to the summer pole." - Done.

Page 1, Abstract, line 13; insert a hyphen between "observation and "hour". - Hyphen added to "observation-hour"; twice more corrections to keep consistency.

Page 1, line 17: use "... has a half-width (full width at half maximum) of roughly 8 km" instead of "has a half width of roughly 4 km". - Corrected half-width as suggested.

Page 1, line 20 and all other occurrences; use "short-wave" instead of "short wave". Use this consistently; see e.g., page 2, line 11, etc. - Corrected all instances of "short wave" to "short-wave"

Page 2, line 14; insert "for" after "investigated" - Done.

Page 2, line 20; clarify the meaning of px. - Exchanged px with pixel for clarification throughout the manuscript.

Page 2, line 24; use "field-of-view" instead of "field of view". - Changed "field of view"

to "field-of-view" twice.

Page 2, line 28; insert a comma after "2015". - Comma added. Also added comma after 2017 on page 2, line 31.

Page 2, line 32/33; "... corresponds to a larger observed area and a lower spatial resolution (70 km x 95 km and 280 m px$^{-1}$ respectively))." instead of "... corresponds to a smaller spatial resolution and a larger observed area (280 m px$^{-1}$ and 70 km x 95 km).". - Changed the order of the sentence as suggested.

Page 3, line 5; insert a comma after "2014". - Done.

Page 4, line 6; explain "Gaussian blur ... with a kernel size of three ... " or provide a reference here. - Gaussian Blur further explained. Added the following sentence and equation after "Additionally, a Gaussian blur is applied to the image with a kernel size of three to reduce high frequency noise.": The kernel (a 3x3 matrix) is calculated with $k(x,y) = \frac{1}{2\pi\sigma^2} \cdot \exp\left(-\frac{x^2+y^2}{2\sigma^2}\right)$; $x, y \in (-1, 1)$ with $\sigma = 1$ and then normalised, so that the sum of all matrix values equals one. The pixels of the smoothed image ($I_{\text{smoothed}}(x,y)$) are calculated as follows, which is commonly called convolution of an image $I$ with a kernel $k$ (Kaehler and Bradski (2017)):

$$I_{\text{smoothed}}(x,y) = \sum_{i=-1}^{1} \sum_{j=-1}^{1} I(x+i, y+j) \cdot k(i,j). \tag{1}$$

Page 6, line 3; "... within the star radius ..." instead of "... with the star radius ...". - "with" corrected to "within" as mentioned.

Page 6, line 16; use "zero-padding" instead of "zero padding" - corrected.

Page 7, line 12; use "... and phase $\varphi$ are determined as follows:" instead of "... and Phase $\varphi$ are determined:" - Added "as follows" as suggested and change "Phase" to "phase".

Page 7, line 23; use "are henceforth denoted as wave events." instead of "... should henceforth be denoted as wave events." - Done.

Page 7, line 24; do you mean "if it occurs more than 30 seconds after the last known signature of the event."? - Reformulated misleading formulation to "A new signature is attributed to a previously identified wave event if it occurs within less than 30 seconds after the last known signature of the (known) wave event."

Page 8, line 5; insert "an" before "important". - Done.

Page 8, line 7; use "seven degrees" instead of "seven degree". - Done.

Page 8, line 9; insert a comma after "filtering". - Done.

Page 11, line 6; the minima occur at 18 UTC and 3 UTC in the NE direction at SBO summer in Figure 5(d)" not at 17 UTC and 5 UTC as stated in the text. This raises a question of the inference of a 12 hour period. - Omitted the statement with the minima and kept it more general: "shows an oscillation-like pattern with a maximum at 23 UTC".

Page 11, line 8; omit "the" after "because". - Omitted by rephrasing the 12h tide issue.

Page 11, Figure 6, yaxis label; correct the spelling of "occurrences"; x-axis label; use "Phase speed (m/s)" instead of "Phasespeed (m/s)" - Corrected y-label to " Occurrences" and x-label to "Phase-speed (m/s)"; also corrected for Fig 7.

Page 12, Table 1 title; replace "in dependence of" by "as a function of" - Done.

Page 13, line 12; use "acoustic-gravity waves" instead of "acoustic gravity waves". - Done.

Page 13, line 19; use "... and northwards as well as southwards at SBO ..." instead of "... and as well northwards as southwards at SBO ...". - Done.

Page 15, line 3; use "constraints on the observations" instead of "restraints for the observations". - Replaced "restraints for" with "constraints on".

Page 15, line 7; replace "Conclusively," by "Therefore," - "Conclusively" replaced by "Therefore".

Page 15, line 18; insert "in the stratosphere" before "(Fleming et al.". - "in the stratosphere" added as suggested for clarification.

Page 16, line 29; The claim of a 12 h period for the NE direction at SBO summer is not supported by the data shown in Figure 5(d) - The specific period of 12h (and also 8h) is omitted.

Please also note the supplement to this comment:
https://www.atmos-meas-tech-discuss.net/amt-2018-322/amt-2018-322-AC1-supplement.pdf
* * *
[Figure]

**Supplement:**

[revised manuscript text omitted]

---

## Author Comment (AC2) · 12 Dec 2018

Reply to Anonymous Referee 2:

We thank Anonymous Referee 2 for the valuable comments from 17th November 2018.

Each comment is considered in the following paragraphs. All changes of the initial manuscript (based on comments of both Anonymous Referees) are tracked in the supplemental material.

Page 1, line 8: "By combining the information of consecutive images". I suggest adding a comma after this part of the sentence to enhance readability. - Comma added as

suggested.

Page 1, line 22: "Thus, the temporal resolution of the FAIM data is comparatively high"
You haven't mentioned yet, what OH bands (i.e. what spectral range) FAIM observes.
Perhaps this should be mentioned, otherwise the reader cannot tell, whether the statement makes sense. - "(mainly OH(3-1) and OH(4-2))" added to the sentence before to give this information at this point.

Page 2, bottom sentences: you give the "average" spatial resolutions for the measurements at the two different sites. It's perhaps interesting for the reader to see, how the spatial resolution varies across the FOV and how it differs between the two different axes. - We provided additional information about the maximum and minimum in resolution for the x- and y-axis, respectively: "The average resolution is calculated by $\sqrt{\frac{A}{N}}$ with the area of the trapezium $A$ and the number of pixels $N$. The resolution of the x-axis $r_x$ (along the top and base side of the trapezium, respectively) ranges from $142\,\mathrm{m\,pixel}^{-1}$ to $199\,\mathrm{m\,pixel}^{-1}$. For the y-axis along the zenith angle, the resolution $r_y$ ranges from $174\,\mathrm{m\,pixel}^{-1}$ to $348\,\mathrm{m\,pixel}^{-1}$ calculated by $r_y = h \cdot (\arctan(\phi_2) - \arctan(\phi_1))$, with the airglow layer height $h$ of $87\,\mathrm{km}$ and $\phi_2$ and $\phi_1$ the elevation angles of two subsequent pixels. The effective pixel size ($\sqrt{r_x \cdot r_y}$) therefore ranges from $157\,\mathrm{m\,pixel}^{-1}$ to $263\,\mathrm{m\,pixel}^{-1}$. Also, see Figure 3 in Hannawald et al. (2016) for the change of average resolution when tilting the instrument along the zenith angle." And later on for SBO: "The x-axis resolution ranges from $164\,\mathrm{m\,pixel}^{-1}$ to $269\,\mathrm{m\,pixel}^{-1}$ and the y-axis resolution ranges from $234\,\mathrm{m\,pixel}^{-1}$ to $636\,\mathrm{m\,pixel}^{-1}$, therefore the effective pixel sizes are $195\,\mathrm{m\,pixel}^{-1}$ to $413\,\mathrm{m\,pixel}^{-1}$."

Page 4, line 5: "where (a) references to the". Perhaps better "where (a) corresponds to the"? - changed according to suggestion.

Page 7, line 2: "that the distance between two maxima has to be at least three pixels (5x5 kernel for maximum search)". Can you explain in a bit more details what "5x5 kernel for maximum search" exactly means in this context? This is not entirely clear, at

least to me. - Changed the respective part "Therefore, a local maximum search is accomplished with the condition that the distance between two maxima has to be at least three pixels (5x5 kernel for maximum search)." to "Therefore, a local maximum search is accomplished: A sliding window of size 5x5 pixels is shifted over the spectrum. If none of the neighbouring pixels are higher than the pixel in the centre of the window, then this pixel is treated as a local maximum."

Page 12, Figure 7, ordinate labels: "Occurences" -> "Occurrences". Same applies to Figure 6. - " labels corrected.

Page 13, line 6: "the standard error calculated with" I don't fully understand what this refers to here? The standard error of what quantity? Is the following mathematical expression correct, i.e. SQRT(6/N)? - The given standard error refers to the skewness values. The skewness can be compared to its standard error and the provided reference states "it is good practice to believe in skewnesses only when they are several or many times as large as this [the standard error]." In the given case the skewnesses are indeed many times as large as their standard errors. To clarify this, we changed "the standard error calculated with $\sqrt{(6/N)}$ (Press et al. (2007)) is 0.04 ..." to "the standard error of the skewness, defined as $\sqrt{\frac{6}{N}}$ (Press et al. (2007)) where $N$ is the number of wave events, is ...".

Page 13, line 12: "which could be related to acoustic gravity waves". These events could also correspond to Doppler-shifted GWs. - We see that this is another explanation for the low periods and reformulated the sentence without the speculation: "Note, that several events have periods smaller than 5 min, but extensive investigation of these events is beyond the scope of this study."

Page 15, bottom paragraph: it would be good to mention what typical meridional wind speeds in summer and winter are. I imagine that the daily mean meridional wind speed associated with the mesospheric residual circulation is quite small. However, tidal variations may be quite large. - We provided additional information from Yuan et al.

2008: "additionally, it is much weaker in winter than in summer (about 10 to $\mathrm{m\,s^{-1}}$ southwards in summer and 0 to $6\,\mathrm{m\,s^{-1}}$ northwards in winter strongly depending on the model and the parameters used; the lidar data at $41°$ N, $105°$ W with tides removed show higher values of up to $18\,\mathrm{m\,s^{-1}}$ in summer and up to $14\,\mathrm{m\,s^{-1}}$ in winter)". Also a short information about the strongest tide, the semi-diurnal tide is added at the last paragraph of discussion: After "However, the 12h solar tide shows the largest change, at least in the zonal wind (**?**)." we added the following sentence: "Conte et al. 2018 showed that the influence of the 12 h solar tide can be up to $40\,\mathrm{m\,s^{-1}}$ for mid-latitudes ($42°$ N)."

Page 16, line 1: Please briefly discuss the suggestions by Vargas et al. (2015) - We shortly added the suggestions of Vargas et al. (2015). "However, there are alternative hypotheses concerning the seasonal dependency of meridional wave propagation. Vargas et al. (2015) lay out that neither the meridional circulation, which is too weak from their point of view, nor tides can explain this seasonal behaviour on a global scale. They suggest an interaction of the seasonally dependent (and strong) zonal wind with the lower thermosphere duct as described by Walterscheid et al. (1999)."

Page 16, line 13 to 18: Please mention that latitude of the Tang et al. observations. - Coordinates for imager data of Tang et al. 2014 added at page 16, line 9: "Results from Tang et al. 2014 (data from imager at $40.7°$ N, $104.9°$ W) coincide with our results."

Please also note the supplement to this comment:
https://www.atmos-meas-tech-discuss.net/amt-2018-322/amt-2018-322-AC2-supplement.pdf